

# Comparative transcriptomics reveals new insights into melatonin-enhanced drought tolerance in naked oat seedlings

Xinjun Zhang[1,*], Wenting Liu[1,*], Yaci Lv[2], Jing Bai[1], Tianliang Li[1], Xiaohong Yang[1], Liantao Liu[3] and Haitao Zhou[1]

[1] Zhangjiakou Academy of Agricultural Sciences, Zhangjiakou, Hebei, China
[2] Hengshui University, Hengshui, Hebei, China
[3] College of Agronomy, Hebei Agricultural University, Baoding, Hebei, China
[*] These authors contributed equally to this work.

## ABSTRACT

The growth and development of naked oat (*Avena nuda* L.) seedlings, a grain recognized as nutritious and healthy, is limited by drought. Melatonin plays a positive role in plants under drought stress. However, its function is unclear in naked oats. This study demonstrated that melatonin enhances drought stress tolerance in oat seedlings. Melatonin application alleviated the declining growth parameters of two naked oat varieties, Huazao No.2 (H2) and Jizhangyou No.15 (J15), under drought stress by increasing the chlorophyll content and photosynthetic rate of leaves. Melatonin pretreatment induced differential gene expression in H2 and J15 under drought stress. Subsequently, the differential gene expression responses to melatonin in the two varieties were further analyzed. The key drought response transcription factors and the regulatory effect of melatonin on drought-related transcription factors were assessed, focusing on genes encoding proteins in the ABA signal transduction pathway, including *PYL*, *PP2C*, *ABF*, *SNRK2*, and *IAA*. Taken together, this study provides new insights into the effect and underlying mechanism of melatonin in alleviating drought stress in naked oat seedlings.

# INTRODUCTION

Drought is among the most widespread abiotic stresses encountered in agriculture and is recognized as a destructive factor impairing plant growth, development, and yield (*Hu & Xiong, 2014*; *Liu et al., 2021*). Within the last decade, due to climate change, poor management of water resources, and intensified competition for limited water resources, the intensity and frequency of drought have increased worldwide (*Berg & Hall, 2017*). As a result, the impact of drought stress has expanded to 64% of arable lands globally and is expected to sharply decline crop yield (*Muthusamy et al., 2016*). Thus, understanding crop responses to plant drought, drought tolerance mechanisms, and drought mitigation measures are crucial to improving plant drought tolerance, agricultural productivity, and global food security (*Meng et al., 2019*).

Corresponding author
Haitao Zhou, zht0206@163.com

One of the aims of plant physiology research is to effectively screen and identify compounds that improve crop stress tolerance. These molecules may also improve plant resistance to abiotic stresses (*Sharma & Zheng, 2019*). Melatonin (*N*-acetyl-5-methoxytryptamine), a phytohormone that regulates plant growth, has anti-oxidant activities, and reduces environmental stress, belongs to this broad class of compounds (*Liu et al., 2021*). Since the discovery of melatonin in plants, its application in agriculture has become an important topic in plant stress resistance research, especially in recent years (*Chen et al., 2021*). Melatonin is broadly involved in the physiological processes of plants and can exert anti-stress effects under adverse conditions, including salinity, drought, cold, high temperatures, and heavy metals (*Chen et al., 2021*; *Debnath et al., 2019*; *Gao et al., 2018*; *Jahan et al., 2019*; *Liu et al., 2021*; *Sharma & Zheng, 2019*). It is a lipophilic and hydrophilic compound that reacts with hydroxyl radicals and peroxyl radicals (*Chen et al., 2021*). As a growth regulator, melatonin also promotes stress resistance of plants growing under adverse conditions such as drought (*Sharma & Zheng, 2019*). During drought stress, melatonin may protect plants from the adverse effects of drought stress by enhancing the efficiency of ROS removal, which helps protect photosynthetic organs and reduces oxidative stress (*Ma et al., 2018*). Moreover, melatonin has also been reported to ultimately enhance the resistance of plants to drought conditions by regulating various physiological, biochemical, and molecular processes (*Chen et al., 2021*).

The mode of action of melatonin in the context of drought response is unclear. Transcriptomic approaches have become an effective tool for systematically examining the action modes of various phytohormones in plants. For example, transcriptome analysis has revealed that the metabolisms of ascorbic acid, aldonic acid, carotenoids, and glutathione help melatonin alleviate drought stress in kiwi seedlings (*Xia et al., 2020*). Drought stress responses mediated by melatonin may involve the activation of calcium signaling by up-regulating the expression of *CNGC*, *CaM*/*CML*, and *CDPK* gene family members in *Davidia involucrata* (*Liu et al., 2021*). The slowing down of chlorophyll degradation after melatonin treatment is due to the down-regulation of genes, including *Chlase*, *PPH*, and *Chl-PRX* (*Ma et al., 2018*). Various key enzymes in the nitrogen fixation pathway are up-regulated by melatonin, including phosphoglycerate kinase (PGK), fructose-bisphosphate aldolase (FBA), transketolase (TKT), and ribulose diphosphate (RUBISCO) (*Liang et al., 2019*). Therefore, melatonin alleviates the harm caused by adverse stress conditions by regulating the expression of transcription factors.

Consuming naked oats (*Avena nuda* L.) reduces fat and sugar levels and the risk of cardiovascular disease in humans; hence, the naked oat is widely recognized as a nutritious and healthy cereal (*Marshall et al., 2013*; *Shewry, 2011*). The protein content of naked oat grains is as high as 15.6%, while its fat content is 3.1–11.6%, much higher than those of wheat, rice, corn, and buckwheat; additionally, it is rich in nutrients such as dietary fibre, flavonoids, and saponins (*Zhao, Bai & Gao, 2017*). Owing to its high energy and protein content, naked oats can be used in livestock feed (*Barneveld, Szarvas & Barr, 1998*). Naked oats are characterized by their drought and cold resistance. However, naked oats are mainly cultivated in arid areas, and drought during the seedling stage is critical environmental stress.

As mentioned above, applying appropriate melatonin concentrations may be a promising strategy for alleviating drought damage to some plant species. However, the involvement of melatonin in drought responses in naked oat remains unknown. The present study investigated the influence of exogenous melatonin on phenotypic and physiological characteristics. Further, melatonin-induced variations in naked oat transcriptome under drought stress were characterized through comparative RNA-Sequencing (RNA-Seq). These results provide a broader understanding of the pathways regulated by melatonin in the naked oat under drought stress.

## MATERIALS & METHODS

### Plant materials and treatments

The experiment was conducted in a smart greenhouse at Zhangjiakou Academy of Agricultural Sciences, Zhangjiakou, Hebei Province, China. This experiment utilized two naked oat varieties: 'Huazao No.2' (H2) and 'Jizhangyou No.15' (J15). The Zhangjiakou Academy of Agricultural Sciences developed both H2 and J15. Naked oat seeds were germinated in an incubator at 25 °C in darkness for 24 h. The germinated seeds were sowed in pots (30 cm long, 18 cm wide, 12 cm high) filled with a 3:1:1 ($v$:$v$:$v$) mixture of topsoil (sampled from the 0–20 cm upper soil layer; organic matter content 10.4 g kg$^{-1}$, total N 1.31 g$^{-1}$, alkali-hydrolyzable N 41.6 mg kg$^{-1}$, available phosphorus 11.5 mg kg$^{-1}$, available potassium 121 mg kg$^{-1}$), nutrient soil (Pindstrup Plus, Ryomgård, Denmark; pH 6.0, screened to 0–6 mm), and sand in an environmentally controlled greenhouse (day/night air temperature, 25/22 °C; photoperiod, 14/12 h; 600 μmol photons m$^{-2}$ s$^{-1}$ light during the day; relative humidity, 60–70%).

Plants of each variety were divided into two groups based on melatonin (MT) application and soil relative water content (SRWC) as follows: well-watered (CK, set as the control group), 75 ± 5% SRWC; drought stress (DS), 45 ± 5% SRWC *Xiao et al. (2020)*; drought stress with melatonin (DS+MT), 45 ± 5% SRWC and sprayed with 100 μM MT (Sigma-Aldrich, MO, USA).

Ten days after emergence, plants were subjected to drought treatments and MT applications. The method of spraying was as follows: all leaves were wet but not dripping; all other treatments received sprayed distilled water; plants were sprayed five times, with each spray given every other day. The SRWC was controlled by the weighing method, and plants were weighed every other day and watered to maintain the appropriate SRWC range throughout the experimentation. Each experiment had nine replicates.

### Measurement of plant growth and soil and plant analyzer development values

The phenotypes of naked oat seedlings were determined starting 5 days after drought treatment (DAD). The aboveground phenotype of naked oats was determined every 10 days, three times. Plant height was measured using a ruler. Leaf area was calculated by the length-width coefficient method. The chlorophyll concentrations of the leaves were determined with a soil and plant analyzer development (SPAD) meter (SPAD-502, Konica-Minolta, Tokyo, Japan).
## Measurement of leaf relative water content

Leaf relative water content (LRWC) in functional leaves was measured at 25 DAD, according to the method described by *Xiao et al. (2020)*. Briefly, the fresh weight (FW) measures were immediately made after sampling. Then, samples were immersed in distilled water for 4 h at room temperature (25 °C) before measuring their turgid weight (TW). The leaf samples were then blotted dry and weighed after being oven-dried at 85 °C for 48 h, after which their dry weight (DW) was measured. The LRWC was calculated based on the following formula: LRWC (%) = [(FW − DW)/(TW − DW)] × 100%.

## Gas exchange and instantaneous water use efficiency

Three representative oat seedlings were selected from each plot for measuring net photosynthetic rate (Pn) and transpiration rates (Tr) using an LI-6400 portable photosynthesis system (LI-COR, NE, USA) under ambient conditions (irradiation = 600 $\mu$mol m$^{-2}$ s$^{-1}$, leaf temperature = 25 ± 1 °C) at 25 DAD. Instantaneous water use efficiency (WUE$_{inst}$) was calculated from the above indicators using the formula: WUE$_{inst}$ = Pn/Tr.

## RNA extraction and cDNA library synthesis

Leaf samples from CK, DS, and DS +MT, were collected at 25 DAD and immediately stored at −80 °C. Each treatment had three biological replicates and no technical replicates. The labeled frozen tissues were ground to a fine powder in liquid N$_2$, and the total RNA was isolated using the TRIzol reagent (Invitrogen, Waltham, MA, USA). The RNA quality was evaluated on a 1% (*w/v*) agarose gel electrophoresis and the RNA Nano 6000 Assay Kit of the Agilent 2100 Bioanalyzer (Agilent Technologies, Santa Clara, CA, USA). The mRNA was extracted from total RNA using poly-T oligo-attached magnetic beads. The mRNA was fragmented using divalent cations under elevated temperature in the First-Strand Synthesis Reaction Buffer (5X). The library fragments were purified with the AMPure XP system (Beckman Coulter, Brea, CA, USA), preferentially selecting 370∼420 bp fragments. The selected fragments were amplified by PCR and purified using AMPure XP beads (Beckman Coulter, Brea, CA, USA) to obtain the final library. The cDNA library was quantified using a Qubit2.0 Fluorometer (Thermo Fisher Scientific, Waltham, MA, USA) and diluted to 1.5 ng/ul. The library insert sizes were determined using Agilent 2100 Bioanalyzer (Agilent Technologies, Santa Clara, CA, USA). After extracting the inserts with preferred sizes (370∼420 bp), a qRT-PCR analysis accurately quantified the library concentration (>2 nM) to ensure quality sequencing of the libraries. Then, high-quality cDNA libraries were pair-end sequenced on the Illumina HiSeq 6000 platform (Illumina, San Diego, CA, USA). The transcriptome datasets are available at the NCBI Sequence Read Archive (SRA), accession numbers SRX12257066–SRX12257083 (Bioproject number PRJNA764537).

## Sequencing read processing and mapping

The high-throughput image data were automatically converted into sequence data (reads) using the CASAVA base recognition software (Illumina, San Diego, CA, USA). Raw reads were cleaned using fastp (version = 0.20.1) with default parameters. First, adapter sequences were trimmed off from all raw reads. Next, undetermined bases (N) were

removed, and low-quality reads (Phred score $\leq 20$ covering $> 50\%$ of the whole read length) were dropped. The Q20, Q30, and GC contents of clean data were calculated by (version = 2.2.1), and clean reads were retained for downstream analysis. After that, the clean reads were *de novo* assembled using the Trinity software (v2.6.6) (*Garber et al., 2011*). The BUSCO software evaluated the splicing quality, accuracy, and completeness of the assemblies from Trinity.fasta, unigene.fa, and cluster.fasta according to the assembly proportion and completeness. Mapping was performed using the FeatureCounts v1.5.0-p3. Filtered Gene Sets were identified, and the resultant read counts were normalized by FPKM (Fragments Per Kilobase of transcript per Million mapped reads) to measure the transcript expression. Gene function was annotated based on the following databases: NCBI non-redundant protein sequences (Nr), NCBI non-redundant nucleotide sequences(Nt), and UniProt (Swiss-Prot) with a $10^{-5}$ e-value threshold.

## Differential expression analysis

Differentially expressed genes (DEGs) between CK, DS, and DS+MT were identified using the DESeq2 R package (1.20.0) via the Novomagic v3.0 platform (https://magic.novogene.com), with a false discovery rate ($p_{adj}$) $< 0.05$ and $\log_2(\text{ratio}) \geq 1$ (*Li et al., 2021*) thresholds for significant differential expression. Enrichment of Gene Ontology (GO) terms was determined using agriGO (https://bio.tools/agrigo), and enrichment of Kyoto Encyclopedia of Genes and Genomes (KEGG) pathways was determined using KOBAS 3.0 (http://kobas.cbi.pku.edu.cn/kobas3/?t=1). The significantly enriched GO items and KEGG pathways with FDR-corrected $p$-value $\leq 0.05$ were analyzed and visualized using OmicShare tools, a free online platform for data analysis (https://www.omicshare.com/tools).

## Quantitative real-time PCR (qRT-PCR) validation

Nine DEGs obtained from the Illumina RNA-seq were randomly selected for qRT-PCR validation with *Actin* as the internal reference gene. Primer3Plus (https://www.primer3plus.com/) was used to create gene-specific primers (Table S1). Each sample had three biological and technical replicates. The quantitative real-time PCR (qRT-PCR) was performed on an ABI 7500 Real-Time PCR System (Applied Biosystems, CA, USA) following the manufacturer's instructions. The amplification program involved: 95 °C for 2 min followed by 45 cycles of 95 °C for 5 s and 60 °C for 1 min. The relative expression level of each unigene was calculated by the $2^{-\Delta\Delta CT}$ method (*Livak & Schmittgen, 2001*). The 7500-system SDS Dissociation Curve Analysis Software examined the amplification specificity of each run.

## Statistical analyses

Statistical analyses for transcriptomic data were performed using three biological replicates and five biological replicates for the other analyses. Analysis of variance (ANOVA) was performed using the SPSS 26.0 software (IBM Corp., Armonk, NY, USA) following Duncan's multiple range tests at the 5% probability level. The results were obtained using SPSS and GraphPad software and presented as mean $\pm$ standard deviation (SD) of three independent biological experiments. The RNASeqPower package in R generated $>0.91$ size estimates for a 2.0 fold change, verifying the high-quality data sets used for differential gene expression analysis.

## RESULTS

### Shoot characteristics under exogenous melatonin

Drought stress significantly affected the growth of the two varieties of naked oat seedlings (Figs. 1A and 1B). Spray application of MT alleviated the effects of drought stress compared with the DS treatment at 25 DAD, especially for H2, which had more upright leaves. In H2, plant height (PH), leaf area (LA), dry weight (DW), and fresh weight (FW) of plants under drought stress decreased by 18.85, 52.35, 43.72, and 57.17%, respectively ($P < 0.05$). However, J15 registered 13.35, 27.03, 32.59, and 45.19% decrease ($P < 0.05$), respectively (Figs. 1C, 1D, 1E, 1E and 1F). The shoot characteristic data show that MT spraying significantly alleviated drought stress in H2, as PH, LA, DW, and FW increased by 9.48, 66.15, 53.42, and 47.6%, respectively, than DS treatments ($P < 0.05$). Similarly, spraying MT improved the shoot characteristics of J15 plants.

### Exogenous melatonin enhances photosynthetic indexes and water use efficiency

Drought stress significantly decreased the Pn and SPAD value in DS-treated leaves ($P < 0.05$) (Figs. 2A and 2B). In H2 plants, the SPAD value and Pn were significantly greater in melatonin-treated than DS treated leaves ($P < 0.05$). However, the MT mitigation effect on J15 was nonsignificant. The LRWC reflection of each treatment was consistent with the SPAD and Pn results (Fig. 2C). The LRWC reflection of each treatment was consistent with the SPAD and Pn data, contrary to $WUE_{inst}$. Drought stress increased the $WUE_{inst}$ of H2 and J15 leaves by 77.03 and 28.16% than CK treatment ($P < 0.05$) (Fig. 2D). MT treatment further increased $WUE_{inst}$, but not significantly.

### Global description of naked oat transcriptome

Eighteen cDNA libraries were prepared from seedlings under CK, DS, and DS + MT treatments to unveil the potential regulatory mechanisms underlying melatonin-mediated drought responses in naked oats. A total of 22.40–27.21 million raw reads were obtained from each cDNA library. After filtering the raw data and checking both the sequencing error rate and GC content distribution, 274.68 Gb of clean reads were obtained, with an average of 15.26 Gb of reads per sample. Per sample, Q20 and Q30 were 97.85 and 93.38%, respectively (Table S2).

Cleaned reads were *de novo* assembled with Trinity, yielding 302,169 transcripts and 95,202 unigenes with 2,110 and 1,239 bp N50 lengths, 726 and 510 bp N90 lengths, respectively (Table S3).

The length distribution of assembled oat transcripts and unigenes revealed that 25.30 and 18.68% of the total transcripts and unigenes were greater than 2,000 bp, respectively (Fig. S1).

All assembled oat unigenes were validated and annotated by implementing similarity searches against several public databases (Fig. S2). Of the 95,202 unigenes, 46.15 and 43.55% aligned to the NR and NT databases. Moreover, 14.65, 35.27, and 7.59% of unigenes exhibited significant matches with the KO, GO, and KOG databases. Ultimately, 57,354 unigenes (60.24%) were annotated in at least one database.

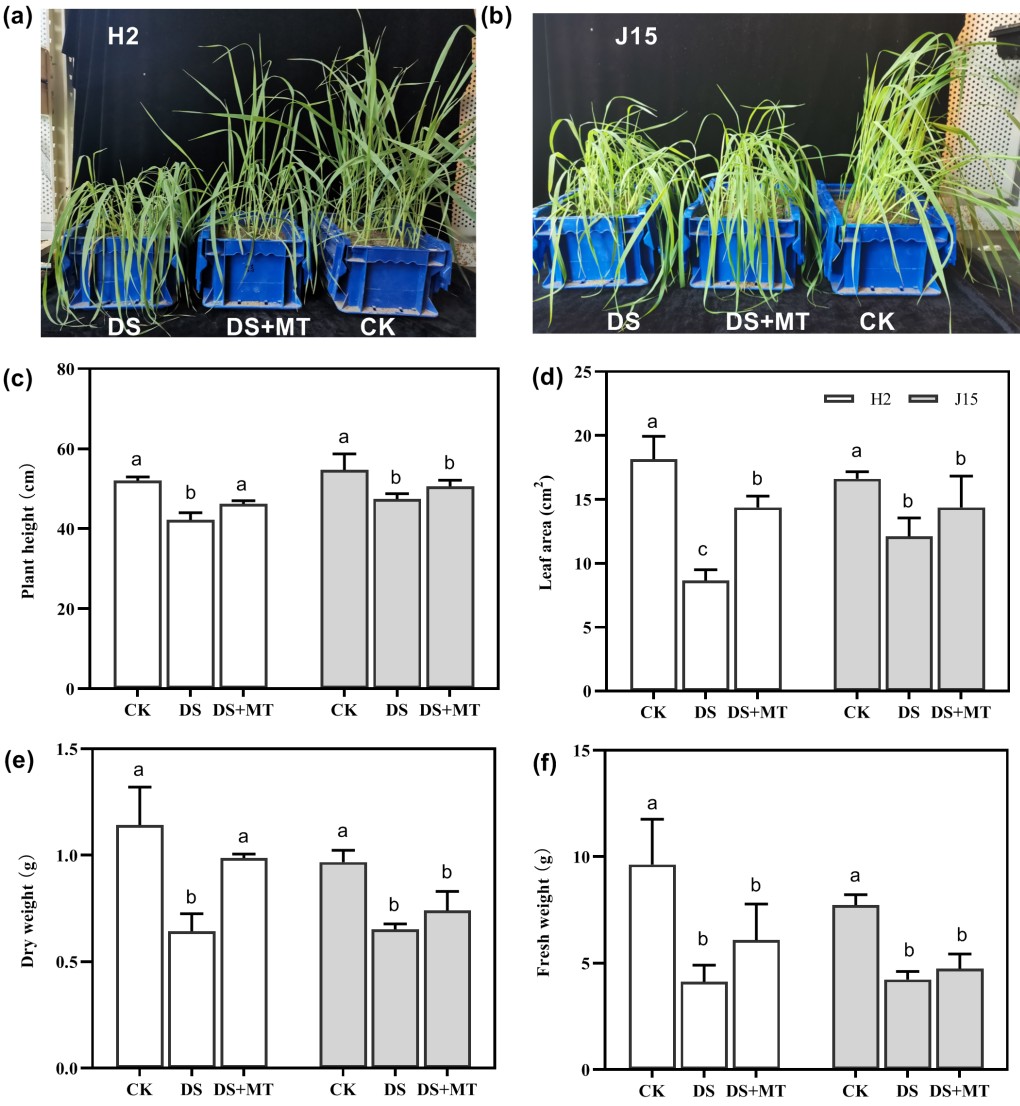

**Figure 1** **Seedlings under well-watered (CK), drought stress (DS), and drought stress with melatonin (DS+MT) treatments at 20 days after drought treatment.** Morphological patterns of H2 (A) and J15 (B). Plant height (C), leaf area (D), dry weight (E), and fresh weight (F) were determined, and the values presented are the mean ± standard error (SE) ($n = 5$). Different lowercase letters within the same plot indicate significant differences between treatment means based on Duncan's multiple range test ($P < 0.05$).

We analyzed the species distribution of the All-Unigene datasets by aligning the sequences with the NR database. A species distribution map was drawn based on the results to characterize the sequence similarity between naked oat and other species (Fig. S3). Naked oat genes were most similar to *Aegilops tauschii* (32.9%). Over 79.3% of the distinct All-Unigenes sequences had top matches with sequences from other Gramineae plant species. Pearson correlation analysis indicated that the three biological replicates had highly consistent transcriptome profiles across all tissue types (Fig. S4).
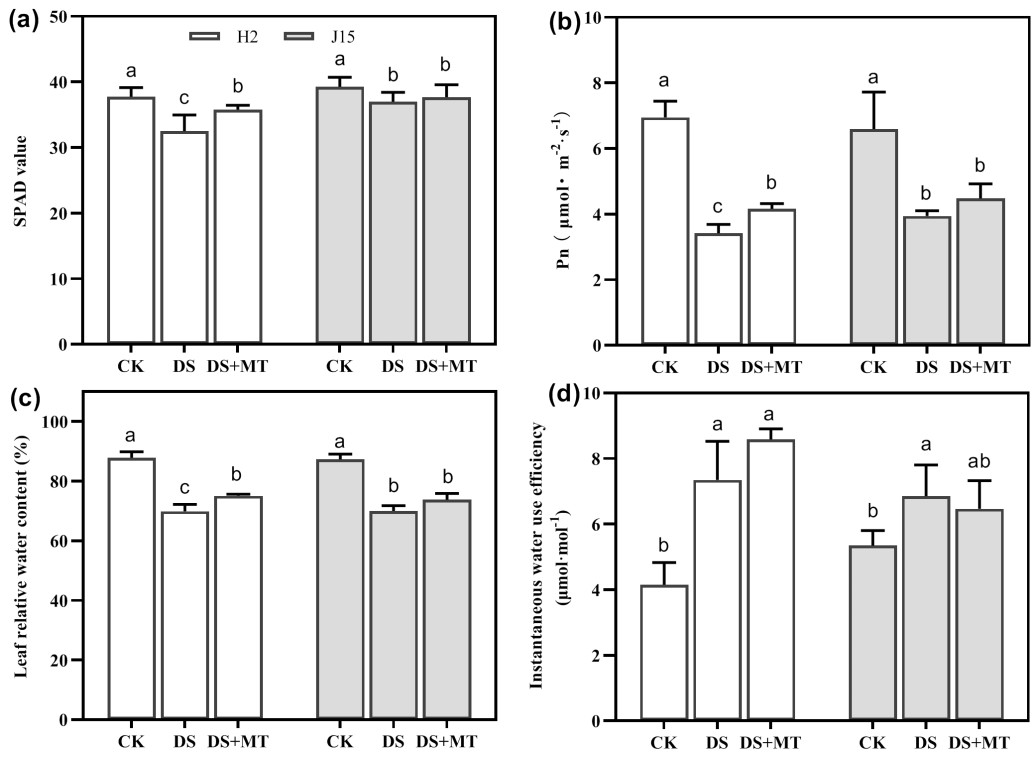

**Figure 2  Melatonin effects on drought stress in leaves of seedling.** Changes in soil and plant analyzer development (SPAD) values (A), net photosynthetic rate (Pn) (B), leaf relative water content (LRWC) (C), and instantaneous water use efficiency (WUE$_{inst}$) (D) of plants under well-watered (CK), drought stress (DS) and drought stress with melatonin (DS+MT) at 20 days after drought treatment (DAD).

## Differential gene expression profiling

We compared the expression profiles of both varieties under CK, DS, and DS+MT treatments at 20 DAD to obtain detailed information about the expression profiles of genes. Uniquely aligned reads were used to estimate gene expression levels as FPKM. As illustrated by the Venn diagram, only significant DEGs were considered for analyzing expression patterns (Fig. 3).

The total DEGs between DS and CK treatments was 4,672 for H2, with 3,586 up-regulated and 1,086 down-regulated (Fig. 3A). For H2 leaves, 4,672 genes were differentially expressed between DS and CK treatments. There were 1,984 DEGs between DS+MT and CK treatments, half the number of DEGs between DS and CK treatments. Between the DS+MT and DS treatments, there were 3,418 DEGs, of which 1,635 were up-regulated and 1,783 were down-regulated. In J15 leaves, 1,160 genes were differentially expressed between DS and CK treatment, 2,661 between DS+MT and CK treatments, and 4,654 between DS+MT and DS, of which 2,073 were up-regulated 2,581 were down-regulated (Fig. 3B). Thus, different varieties responded differently to drought and melatonin treatments. As shown in the Venn diagram (Figs. 3C and 3D), 342 up-regulated and 123 down-regulated genes were observed in H2 and J15 varieties between the DS +MT and CK treatments.
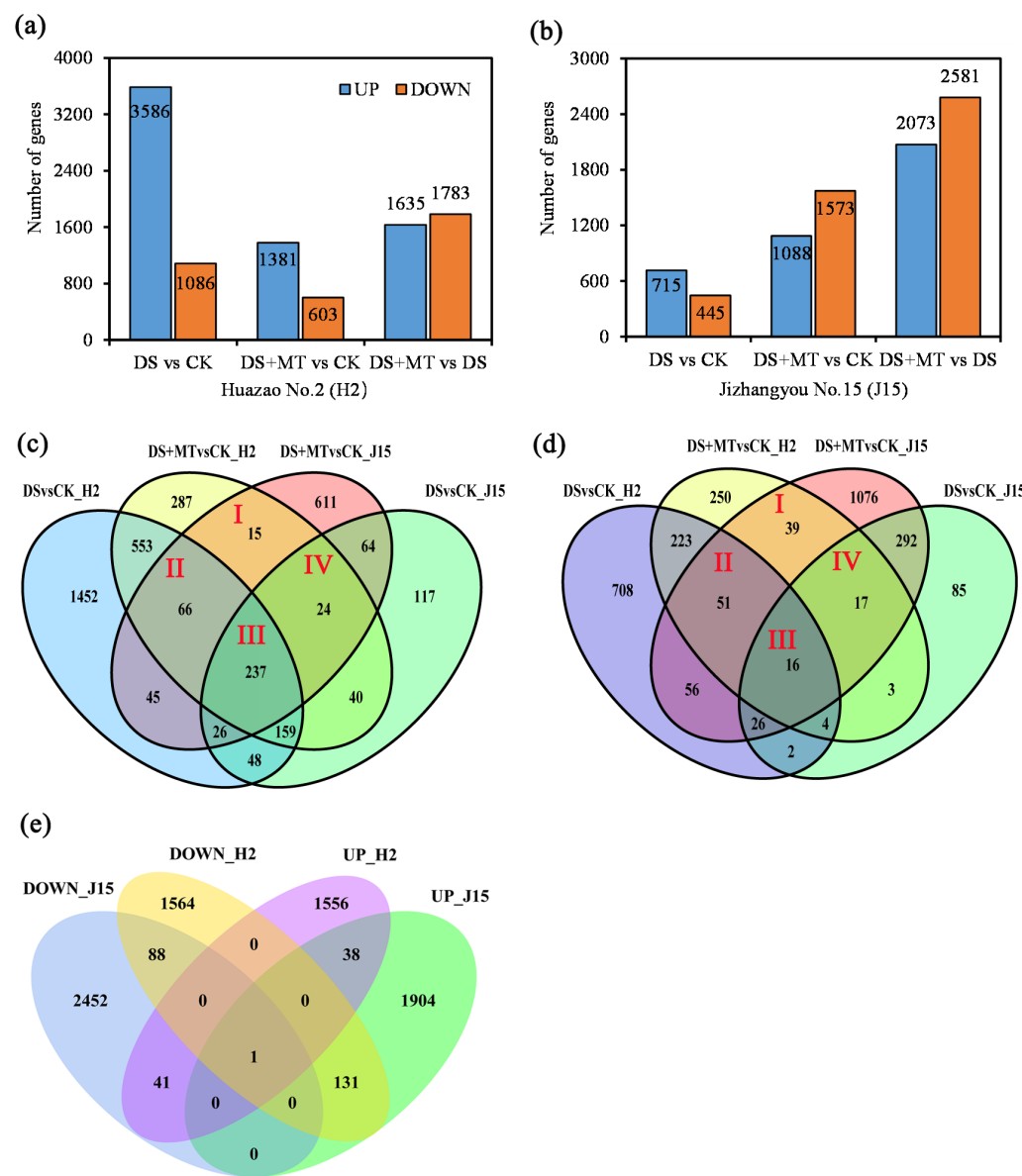

**Figure 3 Bar graphs and Venn diagram of the differentially expressed genes (DEGs) between the DS, DS+MT, and CK treatments.** (A) DEGs upregulated and downregulated in the Huazao N2 (H2) variety. (B) DEGs upregulated and downregulated in the Jizhangyou No.15 (J15) variety. (C) Up-regulated DEGs between DM+MT *vs* CK and DSvsCK. (D) Down-regulated DEGs between DS *vs* CK. (E) DEGs between DM+MT *vs* DS.

The DEGs between DS+MT and DS are presented as the number of co-up-regulated (39) or co-down-regulated (89) genes in both varieties (H2 and J15), respectively (Fig. 3E). Interestingly, there are 172 genes with opposite trends in both varieties, where 41 genes were up-regulated in H2 and down-regulated in J15. Besides, 131 genes were down-regulated in H2 and up-regulated in J15. GO and KEGG analyses were performed for the DEGs between DS+MT and DS.

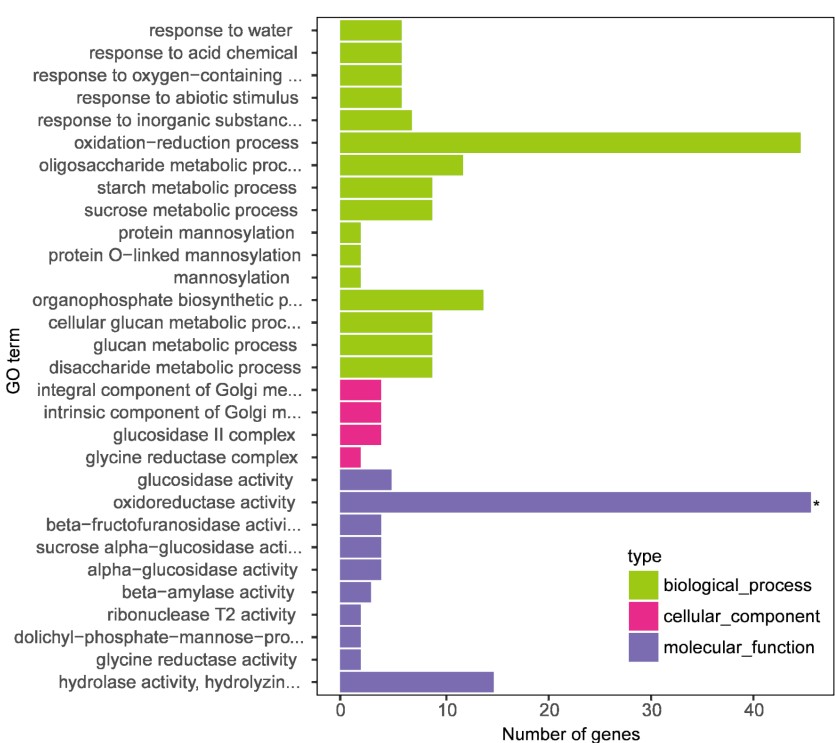

**Figure 4** **Gene Ontology (GO) enrichment analysis of differentially expressed genes (DEGs between DS+MT *vs* DS).** The *x*-axis displays the number of genes, and the *y*-axis indicates the enriched GO terms. The results are summarized according to three major categories: biological process, cellular component, and molecular function.

## Screening and GO classification of melatonin-induced DEGs

GO enrichment analysis was performed on the transcriptome data to understand the role of melatonin supplementation in naked oat seedlings under drought stress. GO terms classified the enrichment of genes into biological processes, molecular function, and cellular components (Fig. 4).

In this research, we focused on the results of two naked oat varieties under drought stress with supplemental melatonin. The results obtained from the GO enrichment analysis with the genes included in each category are summarized in Table S4. Under biological process, the enriched terms included "response to water", "response to acid chemical", "response to oxygen-containing compound", and "response to the abiotic stimulus". For molecular function, the enriched terms included "glucosidase activity" and "oxidoreductase activity". The cellular component included "integral component of Golgi membrane", "intrinsic component of Golgi membrane", "glucosidase II complex", and "glycine reductase complex". These results suggest that leaves of drought-stressed naked oat seedlings underwent global transcriptional reprogramming under supplementation with melatonin.

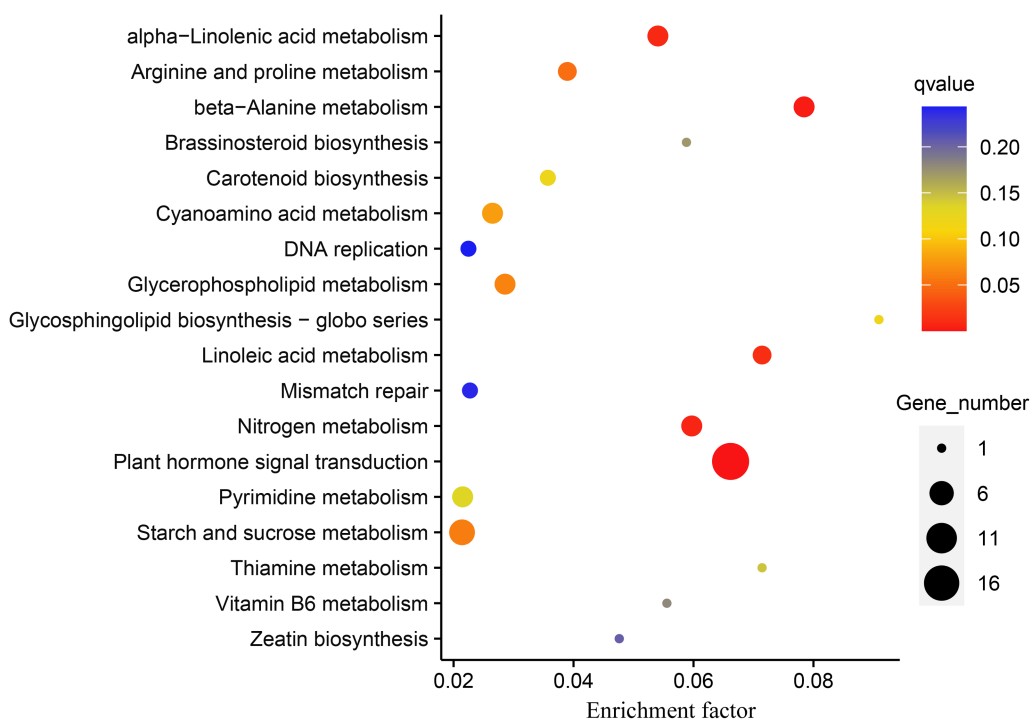

**Figure 5** **Kyoto Encyclopedia of Genes and Genomes (KEGG) pathway enrichment analysis of differentially expressed genes (DEGs between DS+MT *vs* DS).** The number of DEGs in the pathway is proportional to the size of the dots, while the *q*-value is reflected by the color of dots.

## KEGG pathway enrichment analysis

DEGs were subjected to KEGG enrichment analysis to investigate which pathways were involved in melatonin supplementation under drought stress in naked oats. We identified five statistically significant enriched pathways, including "Plant hormone signal transduction", "beta-Alanine metabolism", "Nitrogen metabolism", "alpha-Linolenic acid metabolism", and "Linoleic acid metabolism" (*q*-value < 0.05) (Fig. 5, Table S5).

The "plant hormone signal transduction pathway" (PHSTP) contained the highest number of DEGs, including 16 up-regulated and two down-regulated genes. Among the 16 up-regulated genes, 15 were concentrated in region III of the Venn diagram, and the two down-regulated genes belonged to regions II and III, respectively (Fig. 3). Seventeen DEGs enriched in this pathway were related to abscisic acid (ABA), and one was related to indole acetic acid (IAA) (Table S6). These DEGs observed in the DS + MT treatment were generally expressed at lower levels than under the DS treatment (Table 1). Four valuable down-regulated genes were concentrated in the nitrogen metabolism pathway, including the nitrate reductase, nitrate transporter, and Alpha carbonic anhydrase genes (Table S7). These DEGs observed in the DS + MT treatment were generally expressed at lower levels than under DS treatment.

**Table 1  Comparison of differentially expressed gene enrichment in plant hormone signal transduction pathways between DM+MT *vs* DS.**

| Gene ID | DS+MT_readcount | DS_readcount | DS+MT *vs* DS |
|---|---|---|---|
| Cluster-18670.38166 | 39.65967 | 67.49315 | 0.58761 |
| Cluster-18670.50240 | 1099.53860 | 4925.20922 | 0.22325 |
| Cluster-18670.38545 | 2415.98184 | 2507.68182 | 0.96343 |
| Cluster-18670.32944 | 654.82942 | 1083.28665 | 0.60448 |
| Cluster-18670.24664 | 743.54518 | 1462.97533 | 0.50824 |
| Cluster-18670.34242 | 178.02673 | 668.36947 | 0.26636 |
| Cluster-18670.10980 | 42.25233 | 72.87580 | 0.57979 |
| Cluster-18670.56105 | 256.88457 | 720.62199 | 0.35648 |
| Cluster-18670.40095 | 674.14874 | 3052.11209 | 0.22088 |
| Cluster-18670.57762 | 48.88121 | 363.33726 | 0.13453 |
| Cluster-18670.42950 | 1777.17687 | 4495.30080 | 0.39534 |
| Cluster-18670.43085 | 350.91077 | 751.83223 | 0.46674 |
| Cluster-18670.45168 | 1643.01642 | 2047.72005 | 0.80236 |
| Cluster-18670.45169 | 34.87813 | 51.76962 | 0.67372 |
| Cluster-18670.30764 | 666.19763 | 1670.27340 | 0.39886 |
| Cluster-18670.13862 | 367.92892 | 502.83421 | 0.73171 |
| Cluster-18670.56630 | 454.46697 | 1533.12376 | 0.29643 |

## Expression verification of naked oat seedling DEGs

The reliability of RNA-Seq gene expression was confirmed by qRT-PCR. Nine DEGs consisting of six up-regulated and three down-regulated genes were selected for qRT-PCR analysis (Fig. 6). The fold changes obtained from RNA-seq and qRT-PCR results were highly correlated ($r = 0.998$, $p < 0.01$) (Fig. S5). Hence, the present study's naked oat seedling transcriptome analysis was reliable.

## DISCUSSION

Drought stress is the primary factor limiting plant growth and yield in arid areas, and global climate change is expected to exacerbate these effects (*Shi, Cui & Tian, 2020*). However, effective methods for alleviating drought stress in naked oats are still lacking. Using growth regulators to improve crop drought resistance can help crops overcome this limitation, thus contributing to crop production and global food security (*Yang et al., 2020*). Melatonin is a key regulator that can modulate plant growth and development (*Zhang et al., 2016*). We conducted a transcriptomic study to clarify the molecular mechanism by which melatonin alleviates drought stress in naked oats.

Drought is among the major environmental factors restricting crop growth and yield and affecting the physiology, anatomy, and morphology of crops (*Liu et al., 2015*). Exogenous melatonin induces drought stress tolerance by promoting plant growth (*Muhammad et al., 2021*). In this study, drought stress reduced the growth and development of naked oat seedlings, manifested by a significant decrease in plant height, leaf area, dry weight, and fresh weight. Spraying melatonin alleviated the decline of growth parameters of naked oat seedlings, and PH, LA, DW, and FW values were increased by 9.48%, 66.15%, 53.42%,

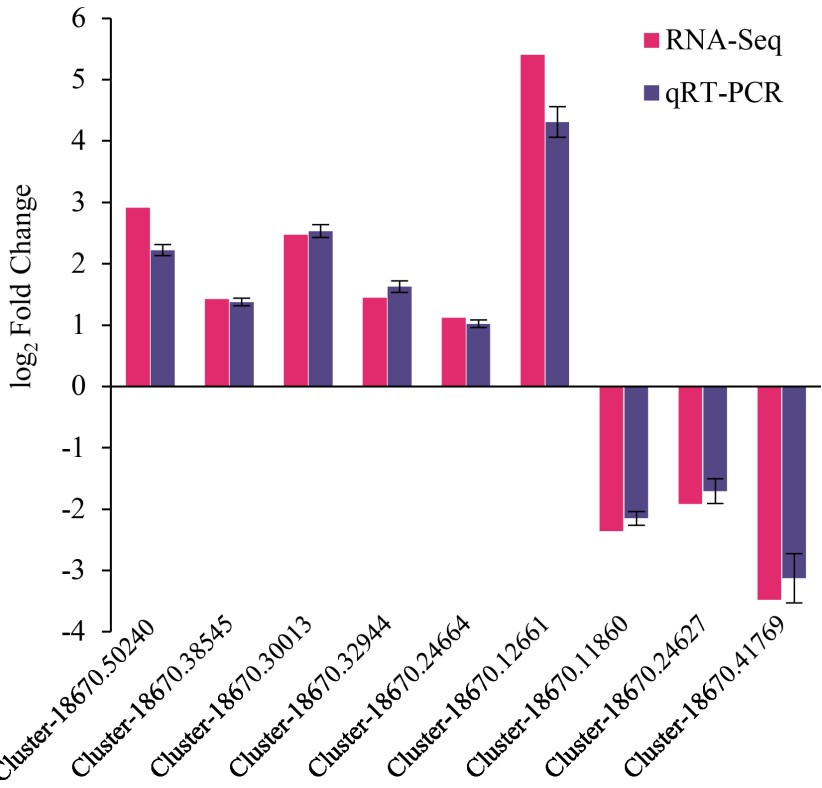

**Figure 6  qRT-PCR validation of differentially expressed genes.** The genes comprised of Cluster-18670.50240 (predicted protein), Cluster-18670.38545 (probable protein phosphatase 2C 9), Cluster-18670.30013 (malonyl-CoA decarboxylase, mitochondrial), Cluster-18670.32944 (protein phosphatase 2C), Cluster-18670.24664 (bZIP transcription factor B), Cluster-18670.12661 (polyamine oxidase-like), Cluster-18670.11860 (alpha carbonic anhydrase 5-like), Cluster-18670.24627 (probable high-affinity nitrate transporter 2.4), Cluster-18670.41769 (nitrate reductase [NAD(P)H]).

and 47.6% compared with DS treatments, respectively (Fig. 1). These results further confirm that melatonin spraying promotes the growth of drought-stressed plants. Spraying melatonin increased the chlorophyll content and photosynthetic rates of leaves under drought stress (Fig. 2).

Similarly, the protective role played by melatonin against drought stress has been observed in various important crops, such as wheat (*Cui et al., 2018*), soybean (*Imran et al., 2021*), maize (*Ye et al., 2016*; *Su et al., 2019*), and cucumber (*Zhang et al., 2013*). Chlorophyll is critical to photosynthesis and plays an important role in transmitting and absorbing light energy (*Arnao & Hernández-Ruiz, 2009*). According to a previous study, adverse conditions significantly reduce the chlorophyll content of leaves, but this could be mitigated via melatonin application. Notably, melatonin can effectively inhibit chlorophyll degradation and thus restore plant photosynthesis (*Shi et al., 2015a*). Similarly, the present study revealed that spraying melatonin enhanced the chlorophyll biosynthesis capacity of naked oat seedlings and increased plant height and biomass. Thus, melatonin potentially induces drought resistance in naked oats throughout arid planting areas.

This study employed transcriptomic analysis to analyze the gene expression of drought-stressed naked oat seedlings treated with MT. The high-quality bases (*i.e.*, Q20 bases) in the sequencing data exceeded 97.85% (Table S2), ensuring the accuracy of subsequent analyses. Moreover, the qRT-PCR validation results for nine DEGs were consistent with the transcriptome analysis, indicating that the transcriptome sequencing results were reliable (Fig. 6). Plant hormones play crucial roles and coordinate diverse signalling pathways during stress responses (*Peleg & Blumwald, 2011*).

The number of DEGs enriched in plant hormone signal transmission pathways was the largest in this study, and they were mainly concentrated in the ABA (PYR/PYL-PP2C-SnRK2-ABF) pathway. Similarly, a previous study reported the substantial role of this signalling pathway in drought stress responses (*Wang, Reiter & Chan, 2018*). Among the DEGs identified in the present study, 11 are members of the Type 2C protein phosphatase (PP2C) gene family (Table 1). The PP2C form the largest family of phosphatases in plants, accounting for 60–65% of all phosphorylases. As a major class of protein phosphatases in plants, PP2C proteins catalyze the dephosphorylation of substrate proteins and regulate signalling pathways, thus participating in various physiological and biochemical processes (*He et al., 2019*). PP2C proteins are vitally involved in ABA signalling (*Park et al., 2009*). ABA induces *PP2C* at the transcriptional level, and PP2C enzymes catalyze reversible phosphorylation (*Liu et al., 2009*). The 11 identified DEGs were all up-regulated under drought stress (Table S6). It agrees with the research conclusion that PP2C is up-regulated under drought stress in tobacco (*Vranová et al., 2000*) and maize (*Xiang et al., 2017*). It is noteworthy that, in this study, spraying melatonin reduced the expression of *PP2C* (Table S7), which likely provides a basis for reducing ABA synthesis and alleviating damage to plants under abiotic stress (*Shi et al., 2015b*). The PP2C gene family in naked oats has not yet been characterized. While a subfamily classification of the 12 PP2C genes isolated in this study is unclear, future studies will fill this gap.

PP2C proteins participate in the ABA signalling pathway by regulating the kinase activity of SnRK or MAPK in response to abiotic stress. In this study, *SnRK2* was up-regulated under drought stress. After spraying melatonin, the degree of up-regulation was alleviated. *SnRK2* is the key gene in the ABA pathway, which negatively regulates ABA synthesis and alleviates drought stress and PP2C (*Umezawa et al., 2004*; *Wang, Reiter & Chan, 2018*). The *SnRK2* gene family has also been identified in soybean and tobacco under drought stress (*Umezawa et al., 2004*). Similar to *PP2C*, in this study, the expression of *SnRK2* decreased after spraying melatonin, which may correspond with *PP2C* inhibiting ABA synthesis. Moreover, the present study demonstrated that the decreased expression of *SnRK2* may not be affected by melatonin concentration. *Mansouri et al. (2021)* also reported a decrease in *SnRK2* in strawberries sprayed with 1,000 µM melatonin.

PYL protein is the abscisic acid receptor and the core component of abscisic acid signal transduction and plays an important role in plant normal growth and development and stress responses (*Wang et al., 2020*). Importantly, PYL plays an important role in plant stress and improving crop yield. For example, PYL9 overexpression has improved drought resistance in *Arabidopsis* (*Zhao et al., 2016*). Additionally, the expression of the abscisic acid receptor PYL4 is strongly down-regulated under stress (*Santiago et al., 2010*). In this

study, the expression of *PYL4* was also strongly down-regulated under drought stress, while spraying melatonin induced the expression of *PYL4* and improved drought resistance. This study revealed that the expression of *IAA* increased under drought stress. Melatonin spraying further induced the expression of *IAA*, which is related to the promotion of root growth, especially lateral root growth, by drought stress, enabling plants to obtain more water. Melatonin may be used as an auxin to promote vegetative growth (*Kolář & Macháčkov, 2010*). Moreover, exogenous melatonin treatment significantly regulated some important genes in plant hormone signalling (*i.e.*, RCAR/PYR/PYL, PP2C, SnRK) (*Haitao et al., 2015*). These results suggest that melatonin regulates the ABA signal transduction pathway (*Fujita, Yoshida & Yamaguchi-Shinozaki, 2013*).

The regulation of the ABA signal transduction pathway by melatonin application is consistent with the findings on physiological indicators. Melatonin spraying regulates ABA synthesis and reduces ABA accumulation, thus increasing SPAD, Pen, and leaf relative water content and finally enhancing stress tolerance.

## CONCLUSIONS

Our study provides new insights into the transcriptomic responses to drought in naked oat seedlings treated with melatonin, the network of multiple hormones, and other molecular responses involved. Both genotypes exhibited a diverse transcriptional response under normal and drought conditions with supplemental melatonin. The key drought response transcription factors and the regulatory effects of melatonin on these transcription factors were assessed, mainly focusing on the genes encoding proteins in the ABA signal transduction pathway, including *PYL, PP2C, ABF, SNRK2*, and *IAA*. Taken together, this study provides new perceptions on the effect and underlying mechanism of melatonin in alleviating drought stress in naked oat seedlings.

## ACKNOWLEDGEMENTS

The authors are grateful to the anonymous reviewers for their valuable comments and suggestions.

### Funding

This work was supported by the Water Physiology and Water Saving Cultivation Post of National Oat and Buckwheat Industry Technology System (CARS-07-B-3), the Sorghum and Oats Post of Hebei Cereals and Beans Innovation Team (HBCT2018070204), the Construction and Demonstration of Oat Organic Base in Zhangbei County (20537501D), and the Scientific and Technological Innovation Team of Modern Seed Industry of Coarse Grains and Beans (21326305D). The funders had no role in study design, data collection and analysis, decision to publish, or preparation of the manuscript.

## Grant Disclosures

The following grant information was disclosed by the authors:

Water Physiology and Water Saving Cultivation Post of National oat and Buckwheat Industry Technology System: CARS-07-B-3.

Sorghum and Oats Post of Hebei Cereals and Beans Innovation Team: HBCT2018070204.

Construction and Demonstration of Oat Organic Base in Zhangbei County: 20537501D.

Scientific and Technological Innovation Team of Modern Seed Industry of Coarse Grains and Beans: 21326305D.

## Competing Interests

The authors declare there are no competing interests.

## Author Contributions

- Xinjun Zhang conceived and designed the experiments, prepared figures and/or tables, authored or reviewed drafts of the article, and approved the final draft.
- Wenting Liu performed the experiments, prepared figures and/or tables, and approved the final draft.
- Yaci Lv analyzed the data, prepared figures and/or tables, and approved the final draft.
- Jing Bai analyzed the data, authored or reviewed drafts of the article, and approved the final draft.
- Tianliang Li performed the experiments, prepared figures and/or tables, and approved the final draft.
- Xiaohong Yang performed the experiments, authored or reviewed drafts of the article, and approved the final draft.
- Liantao Liu analyzed the data, prepared figures and/or tables, and approved the final draft.
- Haitao Zhou conceived and designed the experiments, authored or reviewed drafts of the article, and approved the final draft.

## DNA Deposition

The following information was supplied regarding the deposition of DNA sequences:

All the original sequencing data are avavilable at the NCBI Sequence Read Archive (SRA): PRJNA764537.

## Data Availability

The raw measurements are available in the Supplementary Table.

## Supplemental Information

Supplemental information for this article can be found online at http://dx.doi.org/10.7717/peerj.13669#supplemental-information.

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
