# Peer review of "Comparative transcriptomics reveals new insights into melatonin-enhanced drought tolerance in naked oat seedlings"

_PeerJ, doi:10.7717/peerj.13669_

## Round 0.1 · original submission · Major Revisions

Based on the comments from reviewers, major revision is required for your manuscript.

Reviewer 1 ·

Basic reporting

This study investigated the regulatory mechanism of melatonin in naked oat seedlings under drought stress. The following aspects could be considered for revision.

1.Much more details should be added in the Methods section, most of current methods are too simple, especially for RNA-seq.
2.The correlation between RNA-seq data and physiological analysis should be analyzed and discussed.
3.Most of current discussion are the repeated results, some real discussion based on this study and other published references should be added.

Experimental design

No

Validity of the findings

No

Additional comments

No

·

Basic reporting

This research demonstrated physiological responses and transcriptomic analysis associated with melatonin-regulated drought tolerance in naked oat seedlings.
Major concerns:
1. Trascriptomic data need to be further analysis, and many analyse were unclear. For example, Table 1 only showed basic information about the transcriptome and should be removed to supplemental materials. Table S6 showed comparison of differential gene expression enriched in plant hormone signal transduction pathway between DS + MT and DS, which have been discussed in Discussion section in details associated with melatonin-regulated drought tolerance. So Table S6 should be added in the main body of the manuscript.
2. Please add DEGs between DS+MT and DS (DS+MT vs DS) in the FIg. 3, because the DS+MT vs DS presents the effect of MT in plants under drought stress.
3. Fig. 4 or Fig. 5 showed the GO enrichment analysis or the KEGG pathway enrichment, which was analyzed by using all DEGs? It was unclear in the manuscript. It is better to do the GO and KEGG pathway enrichment using the DEGs between DS+MT vs DS.
4. DEGs in the table 5 and 7 were identified from the DS+MT vs DS? It was unclear.
5. Many unclear or wrong sentences were observed in the Discussion section. For example, line 235-237, and Please delete "being"; add "to" between "crops" and "overcome"; line 241-242; line 252-254;.... The correction and improvement in English should be done throughout the entire paper again.
6. For the Fig. 6, the gene names should be added in this figure. Why were these genes chosen for qTR-PCR validation?

Experimental design

No problems about the experimental design.

Validity of the findings

This research demonstrated the melatonin-regulated drought tolerance in naked oat seedlings associated with plant hormone signal transmission pathways including ABA and IAA pathways.

Additional comments

No additional comments

·

Basic reporting

- Language is clear, unambiguous, and professional
- Sufficient and relevant literature is cited
- Article structure, figures, and tables are professional and properly structured
- Confirmed that raw RNA-Seq data is available in the NIH SRA as indicated in the paper
- Article is self contained with a clear research question and appropriate background to understand the research question and motivation for it
- Results, methods, and conclusions are clearly described and easily understood

Experimental design

- Scope is appropriate
- Research question is clear, well-defined, relevant, and meaningful. Connection to existing literature is made
- Methods are detailed (save one issue, explained below) and clear
- Design of the RNA-Seq experiment is appropriate for the question
- Analysis and interpretation of the RNA-Seq results are clear and reasonable

My one caveat is that the Methods section on the RNA-Seq experiment does not explain how the actual sequencing was done. In other papers, there is usually some text such as:

"mRNA selection, library preparation and sequencing was performed by the Cornell University Life Sciences Core Facilities on an Illumina GAIIx sequencer according to manufacturer specifications. Briefly, mRNA was selected using oligo(dT) probes and then fragmented using divalent cations. cDNA was synthesized using random primers, modified and enriched for attachment to the Illumina flowcell. We sequenced one 60-cycle paired-end lane and two 87-cycle paired-end lanes, generating ∼102.6 million reads for a total of 8,150 MB of sequence." (https://journals.plos.org/plosone/article?id=10.1371/journal.pone.0014202#s2)

"After checking for the quality of RNA with an Agilent 2100 bioanalyzer, samples of nine resistant and nine susceptible mosquitoes were pooled and one RNA-seq library was prepared from each pool. Paired-end Illumina libraries were prepared and run for 80 cycles each end by the Expression Analysis Core at the UC-Davis Genome Center using standard Illumina procedures. Libraries were run at a concentration of 4–5 pM." (https://journals.plos.org/plosone/article?id=10.1371/journal.pone.0044607#s2)

Secondly, there was no discussion of how the sequencing data were processed prior to running DESeq2. How were the data filtered to remove low quality reads? Were they trimmed? How did you do the alignment and/or transcriptome assembly?

I would propose that the authors deposit their generated transcriptomes and associated transcript counts in an archive such as NIH GEO.

Validity of the findings

- Underlying data have been provided
- Statistical analyses are robust and valid
- Conclusions are well-stated, linked to the original research question, and supported by the results

Additional comments

Thank you for the opportunity to read your paper. It was a very interesting study and clear from your results that melatonin could provide to be a valuable resource for drought stress resistance. I appreciate your emphasis on explaining the mechanisms behind its actions. The paper is well written and easy to read.

---

## Round 0.2 · Major Revisions

Please address the concern raised by one reviewer that the authors are now describing completely different analyses of the RNA-Seq data than in previous versions. Further, the descriptions in the results (e.g., using Trinity) do not match the methods (using HiSat2).

Reviewer 1 ·

Basic reporting

OK

Experimental design

OK

Validity of the findings

OK

Additional comments

OK

·

Basic reporting

All my questions have been addressed.

Experimental design

No additional comments

Validity of the findings

No additional comments

Additional comments

No additional comments

·

Basic reporting

- Lines 118 - 122 are very repetitive and could be explained more succinctly

- I was able to access the raw sequencing data in the NCBI SRA using the provided project number.

Experimental design

- The methods are still insufficiently detailed and
- Need to give the database, accession number, genome / gene set versions, and any relevant citations on line 129
- How were the reads cleaned? With what program? What parameters were used?
- Need details on the qRT-PCR experiments beyond just the generation of primers.

- The Results subsection for the transcriptome analysis described a de novo assembly with Trinity. This does NOT match the methods which describes aligning the reads to the genome. Further, Trinity is not cited.

Validity of the findings

- I do not see statistical analyses supporting the conclusions in the first two result subsections. The increase or decrease is maybe less important than whether the differences are statistically significant.

---

## Round 0.3 · Minor Revisions

The Section Editor has commented and said:

"This is a correlational study but the title/abstract/conclusions are written as if the mechanism has been proven. It is demonstrated that MT treatment increases drought tolerance and ABA gene expression, but this is a correlation and does not prove that the change in gene expression is the mechanism by which MT relieves drought stress. To prove this would require testing whether genetic or physiological manipulation of the ABA pathway altered the drought relief caused by MT. The title, abstract, and conclusions need to be rewritten to reflect what is actually shown by this study.

Also: Were FPKM or raw counts used for DEseq?

Figure 1, please have order of plants/treatments the same in the photos and barplots.

Figure 2: mistakes in legend "netphotosyntheticrate" (missing spaces) and "under well-watered (CK), drought stress (DS) and drought" (DS + MT not defined; text cut off?)

Figure 4 what does the asterisk indicate?

Figure 5 "Rich Factor" should be "Enrichment Factor" (check text also)"

·

Basic reporting

No comment

Experimental design

Description of analysis of RNA-Seq data in the Methods is now consistent with the Results. Trinity is mentioned in both places.

Validity of the findings

No comment

---

## Round 0.4 · Minor Revisions

the Section Editor, has commented and said:

"Thank you for the changes. One issue remains: DEseq needs raw counts, not FPKM. See <https://bioconductor.org/packages/devel/bioc/vignettes/DESeq2/inst/doc/DESeq2.html#why-un-normalized-counts>."

Please make change accordingly.

---

## Round 0.5 · accepted · Accept

The author made changes accordingly.